# Tumour-Related Parameters as a Prognostic Factor in Patients with Advanced Cervical Cancer: 20-Year Follow-Up of Diagnostic and Treatment Changes during Chemioradiotherapy

**DOI:** 10.3390/jpm12101722

**Published:** 2022-10-15

**Authors:** Zaneta Warenczak-Florczak, Ewa Burchardt, Agnieszka Wilk, Andrzej Roszak

**Affiliations:** 1Department of Electroradiology, Poznan University of Medical Sciences, 61-701 Poznan, Poland; 2Department of Radiotherapy and Oncological Gynecology, Greater Poland Cancer Centre, 61-866 Poznan, Poland

**Keywords:** cervical cancer, survival, positron-emission tomography, FIGO stage, radiotherapy

## Abstract

Concurrent radiochemotherapy (RCHT) has been the standard treatment for locally advanced cervical cancer since 1999. During this 20-year period, both diagnostic and radiotherapy techniques have developed, such as positron emission tomography (PET) or brachytherapy (BT) planning. The aim of the study was to assess the relationships between prognostic factors and the results of treatment in patients with advanced cervical cancer independent of these changes. The analysis included 266 patients with stage IIB or IIIB FIGO 2009 cervical cancer divided into two groups: one including 147 patients diagnosed with physical examination and ultrasonography (USG) and treated with RCHT with 2D BT from 2001 to 2005; another including 119 patients with metastatic pelvic lymph node diagnosed with PET and treated from 2010 to 2016 with RCHT and 3D BT. The mean five-year overall survival (OS) rate was 59.2% in the first vs. 65.5% in the second group (*p* = 0.048). However, in both groups, stage IIB patients had a significantly higher 5-year OS rate, despite the presence of nodal metastases in group 2. In the first group it was 75.1% in IIB vs. 54.8% in IIIB (*p* = 0.040) 5-year OS and 77.5% vs. 55.8% (*p* = 0.034) in the second group. Important was also a significant association between the dose of BT and survival in group 2: 45.7% vs. 69.2% for dose <28 Gy and 28 Gy (*p* = 0.018). Evolution in the diagnosis and treatment of patients with cervical cancer had led to improvement in the survival of patients and precise treatment with an appropriate stage assessment. However local advance of the tumour is still the most important prognostic factor.

## 1. Introduction

Cervical cancer is one of the most common cancers worldwide. According to GLOBCAN data, there were 604,000 new cases and 342,000 deaths due to cervical cancer in 2020 [1]. Over the past 20 years, both the incidence and mortality rate of cervical cancer have decreased, in part due to early detection and improved diagnostic and therapeutic techniques. In Poland, however, many patients are still diagnosed with advanced cervical cancer, when treatment is less effective. Radical radiotherapy was the treatment of choice until the results of five phase-III randomized trials were published in 1999 showing that combination therapy (radiotherapy plus chemotherapy) was significantly better than radical radiotherapy alone [2,3,4,5,6]. Moreover, advances in diagnostic techniques, most notably positron-emission tomography (PET), have allowed for more precise determination of the disease stage with the detection of metastatic lymph nodes [7]. Similarly, the emergence of more advanced treatment techniques, such as the combination of external beam radiotherapy (EBRT) and brachytherapy (BT), has further improved the treatment outcomes of patients with advanced cervical cancer [8]. PET examination in Greater Poland Cancer Center from 2010 led to a change of treatment method in 30% of patients due to the presence of metastases to the pelvic lymph nodes, paraaortic lymph nodes or distant metastases [9]. Proper staging of cervical cancer leads to personalized treatment of patients and verification of the results in the long term. In this context, the aim of this retrospective study was to evaluate treatment outcomes in patients with stage IIB or IIIB cervical cancer (FIGO 2009 criteria) treated at Greater Poland Cancer Centre over the past 20 years (2001 to 2021), a period during which important changes have taken place in diagnostic and treatment techniques and in staging criteria. We were looking for prognostic factors correlated with 5-year survival rates in two different groups of patients (historical and contemporary) independent of diagnostic and treatment changes made during these 20 years.

## 2. Materials and Methods

We retrospectively evaluated 266 patients diagnosed and treated for stage IIB or IIIB cervical cancer at our centre (Greater Poland Cancer Center, Poznań, Poland) from 2001 to 2021. Follow-up was performed through the end of 2021. The patients were divided into two groups according to the period during which they were diagnosed and treated (2001–2005 vs. 2010–2016). Group 1 consisted of 147 patients with stage IIB (*n* = 32) or IIIB (*n* = 115) disease treated from 2001 to 2005 with follow-up through the end of 2010. Group 2 consisted of 119 patients with stage IIB (*n* = 55) or stage IIIB (*n* = 64) disease treated from 2010 to 2016 and followed until the end of 2021. In both groups, the tumour stage was determined according to FIGO 2009 criteria. We analyzed these two groups independent according to clinical and treatment prognostic factors. Table 1 shows the characteristics of the patients in the two groups.

### 2.1. Patient Characteristics: Group 1

A total of 147 patients diagnosed between 2001 and 2005 with cervical cancer in stage II or III were included in group 1. There were 32 patients in stage IIB and 115 in stage IIIB according to FIGO 2009. Disease progression was determined according to physical examination consisting of per vaginam and per rectum examination, chest X-ray, and ultrasound. Tumour size and the presence of invasion to adjacent structures were assessed. Routine lymph node assessment was not performed. The mean (SD) size of the primary cervical tumour was 4.98 (1.56) cm. For stage IIB, it was 3.96 (1.42), and for IIIB, it was 5.26 (1.48), respectively) (*p* = 0.000). The mean (standard deviation [SD]) age was 51 (8.7) years. Mean body mass index was 25.7 ± 5.1. We observed comorbidities such as hypertension, diabetes and varicose veins of the lower legs in 28.6% of patients.

The histological diagnosis in most of these patients (142/147; 96.6%) was squamous cell carcinoma (SCC). In five cases (3.4%), the histological type was adenocarcinoma. The tumours were graded according to the degree of differentiation: well differentiation—Grade 1 (G1), moderate differentiation—Grade 2 (G2) or poor differentiation—Grade 3 (G3); most of the tumours were classified as G2 (77.5%), followed by G3 (12.3% of cases), and G1 (10.2% of cases). 

All patients were treated with external beam radiotherapy (EBRT) plus brachytherapy and weekly chemotherapy. Conformal EBRT was performed using four photon radiation cross beams (“box” technique), with a dose of 1.8 Gy per fraction. The 4 × 10 cm sheaths from a multi-leaf collimator were used with a dose of 30 Gy to the clinical target volume (CTV). The arrangement was verified on the basis of kilovolt photons. The mean EBRT dose to the tumour was 32.2 (±5.6) Gy, with no significant differences between patients with stage IIB or IIIB disease (*p* = 0.125).

All of the patients in group 1 underwent intracavitary low-dose rate (LDR) brachytherapy, administered with cesium 137 radioisotope and the “afterloading” technique on a Selectron device (Selectron, en ELEKTA company, Stockholm, Sweden). Treatment was delivered in two fractions. The dose rate at the reference point was 1.0–1.2 Gy. Planning of the dose distribution in the applicators was based on x-rays and virtual planning in accordance with International Committee for Radiological Units 38 (ICRU 38) recommendations [10]—Figure 1. The mean brachytherapy dose to the tumour was 52.2 (±8.6) Gy with no significant differences in doses between stage IIB and stage IIIB patients (*p* = 0.115).

The total dose for the combined treatments in group 1 was 87.1 (±10) Gy, with no significant differences between patients with stage IIB or IIIB disease (*p* = 0.164). 

All patients also received adjuvant chemotherapy with Cisplatin with a dose of 40 mg/m^2^ administered once weekly during EBRT. The mean number of courses was 4.73 ± 1.1, with no differences between the two stages (stage IIB and IIIB).

### 2.2. Patient Characteristics: Group 2

Group 2 included 119 patients treated from 2010 to 2016. There were 55 patients in stage IIB and 64 in stage IIIB according to FIGO 2009. The diagnostic work-up was similar to that applied in group 1 (i.e., physical examination with per vaginam and rectal examination), but with the addition of fluorine-18 deoxyglucose (18FDG) PET imaging to check for the presence of nodal invasion (pelvic and/or para-aortic lymph nodes) and/or distant metastases. As a result of the analysis, we included only patients with pelvic lymph node metastases, obtaining a relatively homogeneous group. According to FIGO 2009 criteria, all of these patients were classified as stage IIB or IIIB, despite the presence of pelvic node disease. If the revised 2018 FIGO criteria had been in place at the time of diagnosis, these patients would have been considered stage IIIC1. The mean (SD) number of metastatic lymph nodes in group 2 (based on PET imaging) was 2.42 (±1.59). In patients with IIB, the mean number of involved nodes was 2.44 (±1.49) and 2.42 (±1.59) in the IIIB stage, respectively (*p* = 0.616). The mean (SD) size of the primary cervical tumour in the PET was 5.34 (1.48) cm. For stage IIB, it was 4.89 (1.51) and for IIIB it was 5.56 (1.22), respectively (*p* = 0.002).

The mean (SD) age of these patients was 52.7 ± 10.7 years. The mean body mass index was 26.9 ± 4.9. We observed comorbidities such as hypertension, diabetes and varicose veins of the lower legs in 14.2% of patients.

The histopathologic classification for most of these tumours (85.7%) was SCC, with the remaining 14.3% classified as adenocarcinoma. The differentiation status was as follows: well differentiation (G1) in 3.4% of patients, moderate differentiation (G2) in 74% of patients and poor differentiation (G3) in 22.6%. 

All patients in group 2 received the same treatment as in group 1 (EBRT plus brachytherapy and weekly chemotherapy). Most patients (*n* = 112, 94.1%) underwent three-dimensional conformal radiotherapy (3D-CRT) delivered through a linear accelerator equipped with a multileaf collimator (MLC)( Varian Medical Systems, Palo Alto, CA, USA) which enabled mapping of the irradiation field in accordance with the recommendations of ICRU reports 50 and 62. The total dose was 50.4 Gy (whole pelvis plus the tumour) with 1.8 Gy per fraction. In the remaining seven patients (5.9%), the radiation modality was IMRT, due to anatomy that requires a change in the arrangement of radiation beams. Patient positioning on the treatment table was regularly verified through kilovolt and megavolt imaging.

In the second group, patients were also divided into two subgroups (boost and no boost subgroup), depending on received a boost to the metastatic lymph nodes diagnosed in the PET. A 10 Gy boost (IMRT) was delivered in 66 patients (55.5%). In this subgroup, 28 patients were stage IIB (50.9%) and 38 were stage IIIB (59.4%).

In the second group, patients were treated with high-dose-rate (HDR) intracavitary brachytherapy with the use of Fletcher or Ring applicators(Elekta Brachytherapy, ELEKTA company, Stockholm, Sweden). The brachytherapy dose (>12 Gy/h) was delivered with the Nucletron device(Nucletron, en ELEKTA company, Stockholm, Sweden). Computed tomography (CT) or magnetic resonance imaging (MRI) were used to determine the gross tumour volume (GTV). MRI was also used to check for the presence of invasive disease and to determine the high-risk clinical target volume (HR-CTV), including the cervix, vaginal vaults, and part of the uterine body. A total dose of 28 Gy was prescribed to cover 90% of the HR-CTV administered in four fractions based on 3D imaging (CT or MRI) according to The Groupe Européen de Curiethérapie (GEC) and the European Society for Radiotherapy and Oncology (GEC-ESTRO) working group protocol [11]—Figure 2. The mean brachytherapy dose was 27.7 Gy.

The patients also received chemotherapy—Cisplatin with 40 mg/m^2^ administered once weekly for the entire duration of EBRT. The mean (SD) number of chemotherapy courses was 3.67 ± 1.96, with no differences between the two stages (stage IIB and IIIB).

### 2.3. Statistical Analysis

All calculations were performed with the statistical package STATISTICA v.10 (StatSoft Inc, Neshville, USA) [12]. The cut-off for statistical significance was set at *p* = 0.05. The Pearson Chi-square test, the Fisher test, and the Mann–Whitney test were applied as appropriate. The Kaplan–Meier algorithm was used to construct the survival curves. Kaplan–Meier curves were compared using the log-rank test.

## 3. Results

The mean five-year overall survival (OS) rate was 59.2% in group 1 versus 65.5% in group 2, a statistically significant difference of 6.3 percentage points (*p* = 0.048). The results are presented in Figure 3.

The mean (SD) survival time of the patients who died during follow-up was 1.9 ± 0.9 in the first group and 2.6 ± 1.3 years in the second group, respectively.

### 3.1. Influence of Clinical-Related Factors on Outcomes in the Groups

In both groups, stage IIB patients had a significantly higher 5-year OS rate, despite the presence of nodal metastases in group 2. In the first group, the 5-year OS rate was observed in 75.1% of patients in stage IIB and in 54.8% of patients in stage IIIB with statistical significance (*p* = 0.040). In the second group, the corresponding 5-year OS rates were observed in 77.5% of patients with IIB stage and in 55.8% of patients with IIIB, respectively (*p* = 0.034). The results are presented in Figure 4.

The variable most strongly correlated with 5-year survival outcomes was the clinical stage. Tumour size (measured by ultrasound in the first group and PET in the second one) was also correlated with survival. In group 1, the mean (SD) tumour size of patients alive after 5 years of follow-up was 4.7 (±1.5) vs. 5.3 (±1.6) cm in those who died during this time period (*p* = 0.015). In group 2, the mean (SD) tumour size of patients alive after 5 years of follow-up was 4.9 (±1.2) vs. 5.4 (±1.4) cm in those who died during this time period (*p* = 0.003).

Neither the histopathological classification (SCC or adenocarcinoma) nor the differentiation grade was significantly associated with 5-year OS rates in either group, with *p* = 0.831 in the first group and *p* = 0.537 in the second one.

In group 2, in which all patients had metastases to pelvic lymph nodes, we found no statistically significant association between the number of metastatic nodes and survival outcomes. We observed that the 5-year survival rate was similar in patients with one metastatic lymph node and three and five or more metastatic lymph nodes (*p* = 0.677). The results are presented in Table 2.

### 3.2. Influence of Treatment-Related Factors on Outcomes in the Groups

The EBRT dose to the pelvis had no significant effect on 5-year OS in either group. In the first group: *p* = 0.317 and in the second one: *p* = 0.281.

Analysis of the brachytherapy dose shows that there was no association with OS in the first group 1 (*p* = 0.711). There was, however, a significant association in the second group. In patients who received a dose lower than 28 Gy, the 5-year OS was observed in 45.7% of them. In patients who received doses of 28 Gy, we observed 5-year OS in 69.2% of them (*p* = 0.018). The results are presented in Figure 5.

There was no correlation between the total tumour dose and OS, regardless of the clinical stage in either group (*p* = 0.831 and *p* = 0.376 in groups 1 and 2, respectively). All patients received a total dose >85 Gy.

All patients included in group 2 had metastatic lymph nodes. Of the 119 patients in this group, 66 patients (55%) received a 10 Gy boost by IMRT to the metastatic nodes. OS at 5 years in the boosted group was observed in 64.9% of patients versus 65.4% in the non-boost group of patients, named as the control subgroup (*p* = 0.903). There were also no significant differences in 5-year OS observed in patients who received a boost versus no boost, regardless of disease stage. In patients with stage IIB, it was 78.6% vs. 77.4% (*p* = 0.96) and in stage IIIB patients: 57.5% vs. 53.1% (*p* = 0.708). The results are presented in Figure 6.

Finally, the number of courses of chemotherapy had no significant influence on OS rates in either group (*p* = 0.419 and *p* = 0.821 in groups 1 and 2).

## 4. Discussion

We performed this study to evaluate and compare treatment outcomes and predicting factors in patients with stage IIB or IIIB cervical cancer (FIGO 2009 criteria) treated at our institution during two distinct time periods (2001–2005 and 2006–2010) over the past 20 years of changes in diagnostic and treatment methods. The mean five-year OS rate was significantly higher in group 2 (65.5%% vs. 59.2%) (Figure 3), despite the fact that all of the patients in group 2 had pelvic node involvement (and thus would have been classified with stage IIIC disease according to the FIGO 2018 criteria). Numerous studies have analyzed the patient and treatment-related variables that have the greatest impact on survival, including the proper diagnosis and assessment of the disease stage, radiation dose rates (both from EBRT and brachytherapy) and the role of modern radiotherapy techniques (3DCRT, IMRT) and brachytherapy (various radioactive sources and 3D techniques) [8,9,11,13,14]. This difference in survival in our study, despite the high clinical stage, is mainly attributable to the use of PET-CT in the diagnosis, which not only detected the presence of pelvic node metastases but also led to higher staging for patients with para-aortic node and/or distant metastases, who received a different treatment protocol (and were therefore not included in this study). By contrast, the pre-treatment diagnostic imaging in group 1 consisted of x-ray and ultrasound examination, which are only capable of providing limited information about the patients’ true disease status. In this regard, it is highly likely that some of the patients in group 1 had pelvic node metastases and possible other metastatic lesions outside the target area. However, these metastases were not detected due to the inherent limitations of these less sensitive and less specific imaging modalities.

Chemoradiotherapy has been the standard of care for patients with advanced cervical cancer from 1999 until now [13,14]. Since that time, substantial changes in diagnostic imaging and improved treatment approaches have improved survival outcomes in these patients. However, it is important to determine the factors that have the greatest influence on survival independent of these changes. In our sample, the combination of radiotherapy and chemotherapy produced good results (5-year OS rates of 59.2% in the first group and 65.5% in the second, respectively, as shown in Figure 3). These results are comparable to those obtained in other studies evaluating the effectiveness of the combination treatment. For example, 5-year OS rates in the NCIC (Nationale Cancer Institute of Canada) and RTOG (Radiation Therapy Oncology Group) 90–01 studies involving patients with stage 1B to IIIB disease were 63–66% [4,15]. 

In our study, we were looking for prognostic factors mostly correlated with survival in these two groups of patients (modern and historical) independent of changes in diagnostic and treatment methods mentioned in the first place as associated with better survival in the modern group. In our analysis, the variable that had the greatest impact on survival was the clinical stage (2009 FIGO criteria), in line with previous reports [16,17,18,19,20], as evidenced by the large difference in 5-year survival rates between patients with stage IIB vs. IIIB disease: 75.1% vs. 54.8% in group 1, and 77.5% vs. 55.8% in group 2 (Figure 4). 

The size of the primary tumour is also associated with the clinical stage and treatment outcomes [21,22,23,24]. In our study, 5-year OS was inversely related to tumour size and the primary tumour was significantly larger in patients with stage IIIB disease in both groups of patients. 

The presence of lymph node metastases in locally advanced cervical cancer influences therapeutic decision-making, including the radiotherapy dose and chemotherapy regimen. In this regard, the value of 18FDG-PET-CT in the diagnosis of nodal metastases in patients with cervical cancer has been well-established and the availability or lack thereof of this diagnostic technique is known to influence survival outcomes. Surprisingly, 18FDG-PET-CT is not considered a standard imaging test in the diagnostic work-up of cervical cancer. In a 2010 study, Kidd et al. demonstrated that the presence of lymph node metastases on PET-CT scans was associated with a worse prognosis at all stages of cancer advancement [25]. In that study, pelvic node metastases were detected by PET-CT, with a significant difference in 3-year OS outcomes: 58% vs. 73% for patients with and without nodal metastases on the PET-CT scan, respectively. In our study in group 2, with metastatic lymph nodes, regardless of the disease stage, there was no statistically significant relationship between the number of affected lymph nodes (one, two or more) and OS (*p* = 0.677) (Table 2).

In our study, all patients in group 2 underwent PET-CT imaging, which allowed for the assessment not only of local tumour progression but also the determination of the presence of metastases to the lymph nodes and neighbouring organs. The introduction of the new FIGO classification in 2018 [26], in which patients with N1 disease were upstaged to stage IIIC1 disease, provides a simpler and clearer path to treatment selection [27]. However, in a study published in 2019, Wright et al. observed that patients with metastatic stage IIIC1 disease comprise a highly heterogeneous group [28]. Similarly, in our patients, we found that survival outcomes were mainly dependent on the local tumour stage rather than on the presence or not of nodal metastasis in the second group.

The radiotherapy dose depends on tumour size and local advancement. Other studies also have demonstrated the prognostic significance of the radiation dose [29,30]. Both EBRT and brachytherapy allow for the administration of high doses without increasing toxicity to healthy organs. There is a clear association between the total radiotherapy dose to the tumour and treatment outcomes. However, we found no correlation between the mean total dose and 5-year OS in either group, probably due to the relatively high mean dose (>85 Gy), which was consistent with the ICRU recommendations. However, in the patients in group 2 who were treated with advanced 3D-HDR brachytherapy, the dose was significantly correlated with survival outcomes. In some of these patients, we were unable to administer the full dose due to patient anatomy and/or the tumour location (adjacent to healthy organs) or size. As a result, the OS rate was lower in patients who received a brachytherapy dose <28 Gy versus those who received a dose ≥28 Gy (45.7% vs. 69.2%) (Figure 5). Combining intracavitary techniques with intra-tissue brachytherapy improves treatment outcomes. Given the much greater precision in imaging and treatment planning achievable with 3D brachytherapy, proper tumour assessment is essential, as is the selection of the most appropriate applicators based on the needs of the individual patient [31,32].

## 5. Conclusions

The diagnosis and treatment of advanced cervical cancer have improved substantially in the last 20 years, leading to better survival outcomes. The introduction of PET-CT imaging as part of the diagnostic process allows clinicians to better assess local disease extension and to identify metastases to the lymph nodes and distant organs.

Radiotherapy techniques have also improved substantially during this time period, and HDR brachytherapy is increasingly used in 3D planning, thus allowing for full radiation coverage of the whole tumour volume with better protection of healthy organs and highly precise dose delivery.

However, the results of the present study, which involved patients with advanced cervical cancer treated in two distinct time periods (2001–2005 and 2006–2010), show that local tumour advancement remains the most important parameter for survival at 5 years. This finding underscores the importance of cancer prevention and early detection to ensure the best outcomes in patients with cervical cancer.

## Figures and Tables

**Figure 1 jpm-12-01722-f001:**
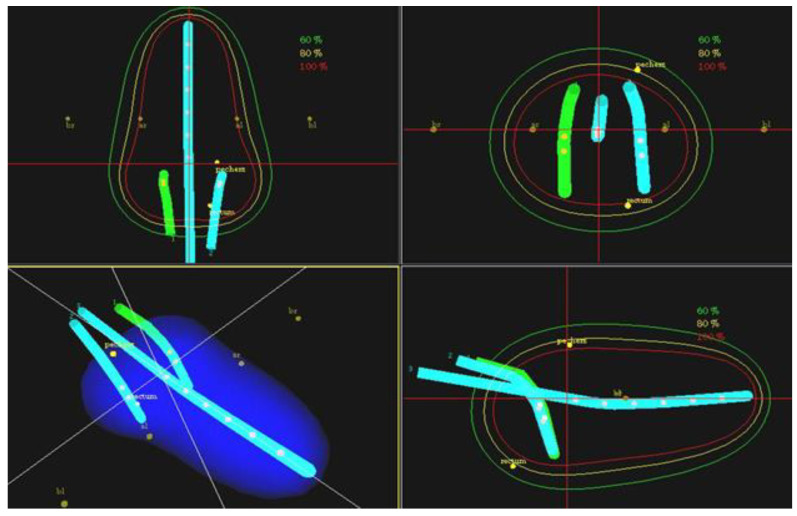
LDR brachytherapy planning with 2D X-rays—virtual plan to points A in three dimensions.

**Figure 2 jpm-12-01722-f002:**
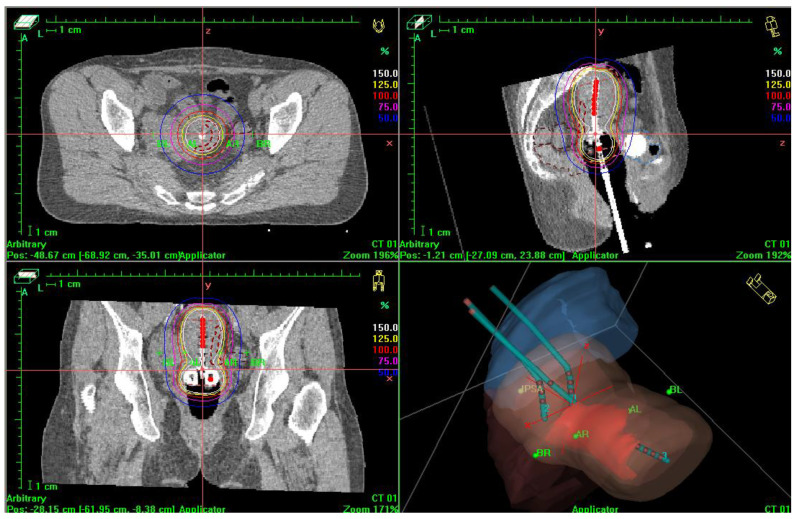
Three—dimensional planning in HDR brachytherapy with CT scans—plan to HR-CTV in three dimensions.

**Figure 3 jpm-12-01722-f003:**
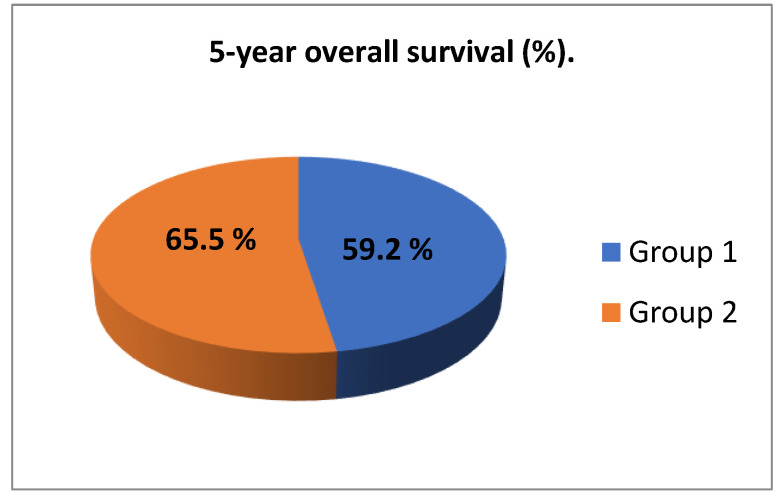
Five-year overall survival (OS) rate in the two analysed study groups.

**Figure 4 jpm-12-01722-f004:**
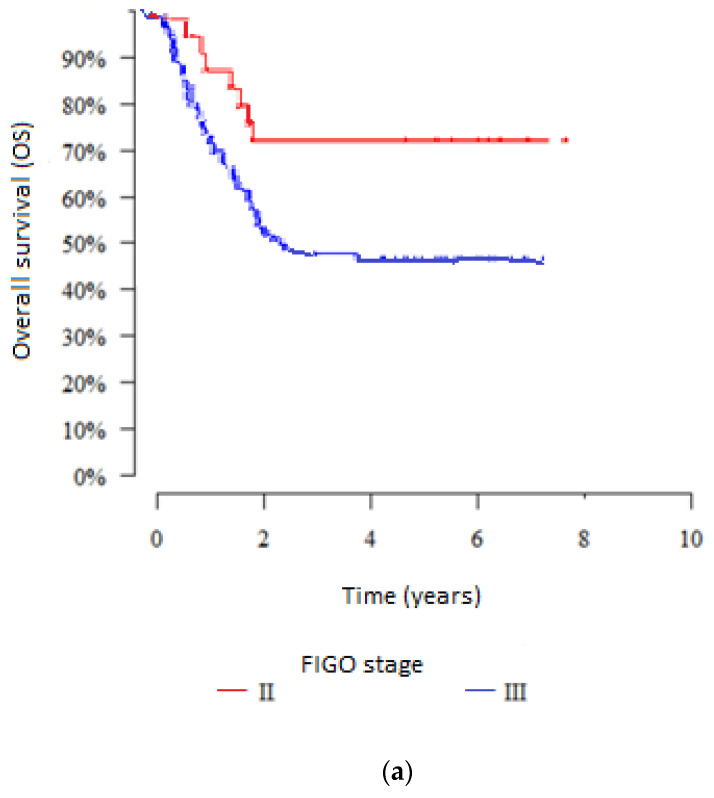
Comparison of 5-year OS rates between stage IIB and IIIB patients in group 1 (**a**) and group 2 (**b**).

**Figure 5 jpm-12-01722-f005:**
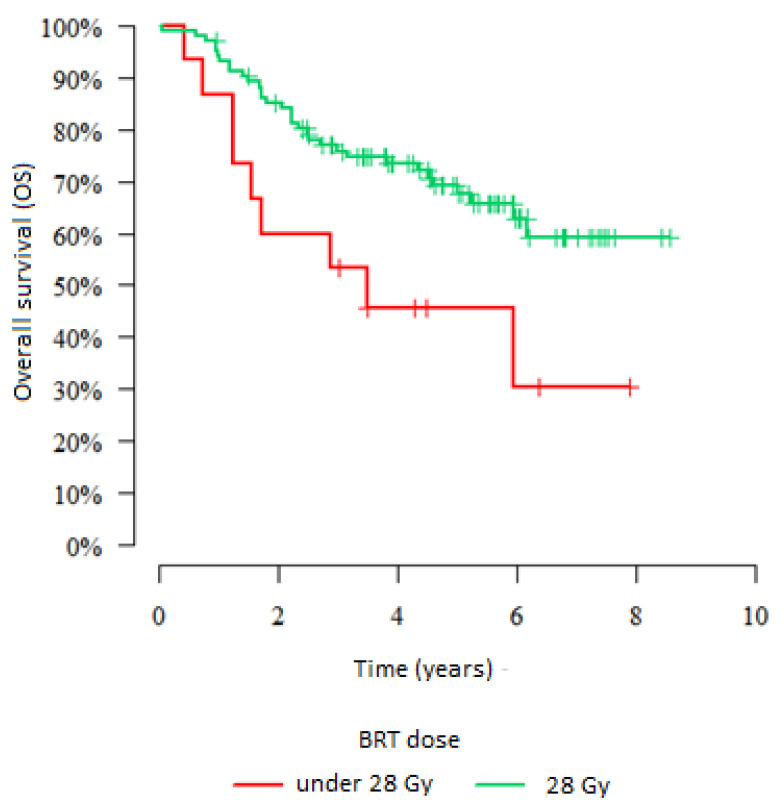
Association between brachytherapy dose lower than 28 Gy or 28 Gy and 5-year overall survival.

**Figure 6 jpm-12-01722-f006:**
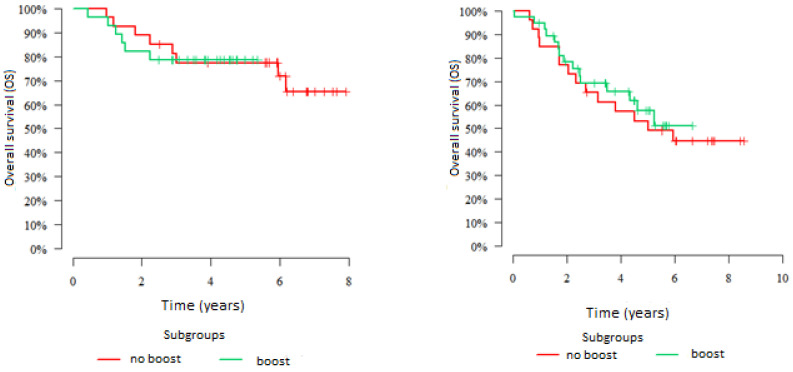
Five-year overall survival of patients in group 2 according to boost status (IMRT, 10 Gy; yes vs. no). Patients with stage IIB disease are shown on the **left** with stage IIIB on the **right**.

**Table 1 jpm-12-01722-t001:** Clinical and treatment-related characteristics of groups 1 and 2.

Characteristics		Group 1, *n* = 147	Group 2, *n* = 119
Treatment Period		2001–2005	2010–2016
Disease stage			
Stage IIB, n		32	55
Stage IIIB, n		115	64
Tumour size in cm (SD)		4.98 (1.56)	5.34 (1.48)
Mean (SD) age, years		51 (8.7)	52.9 (10.7)
Comorbidities disease, %		28.6	14.2
Mean BMI (SD)		25.7 (5.1)	26.9 (4.9)
Histopathologic diagnosis	Squamous cell ca.	96.6%	85.7%
	Adenocarcinoma	3.4%	14.3%
Diagnostic technique		X-ray, USG	PET-CT
EBRT technique		3D-CRT	3D-CRT/ IMRT
Brachytherapy	Dose rate	LDR	HDR
	Technique	2D	3D
EBRT dose (SD)		32.2 (5.6)	48.6 (1.8)
Brachytherapy dose (SD)		52.2 (8.6)	38.9 (0.9)
Mean (SD) number of chemotherapy courses		4.73 (1.1)	3.67 (1.96)
Follow-up period, years		5	5

Abbreviations: SD, standard deviation; BMI—Body Mass Index; LDR, low-dose-rate; HDR, high-dose-rate; 2D, two-dimensional; 3D, three-dimensional. EBRT, external beam radiotherapy. IMRT, intensity-modulated radiotherapy. PET-CT, positron-emission tomography/computed tomography; US, ultrasound.

**Table 2 jpm-12-01722-t002:** Association between the number of metastatic lymph nodes and 5-year OS in group 2.

Number of Metastatic Lymph Nodes	Number of Patients	Overall Survival	
		5-Years (%)	*p* *
1	34	70.47%	*p* = 0.677
2	41	63.35%	
3	21	72.43%	
4	12	55.56%	
≥5	10	60.00%	

* *p*—test LR (log-rank).

## Data Availability

Not applicable.

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
