# Peer review of "Tumour-Related Parameters as a Prognostic Factor in Patients with Advanced Cervical Cancer: 20-Year Follow-Up of Diagnostic and Treatment Changes during Chemioradiotherapy"

_jpm, 2022, doi:10.3390/jpm12101722_

Round 1

Reviewer 1 Report

This manuscript examined the differences in results of radical chemoradiotherapy for cervical cancer between historical periods. The reasons for the better outcomes with the newer treatment periods are probably a combination of a variety of factors, including diagnostic accuracy for pelvic lymph node metastases, improved accuracy of external-beam irradiation (boost irradiation with IMRT), improved brachytherapy (from 2-D to 3-D), and changes from low-dose rate to high-dose-rate.

The scientific hypothesis of this study is blurred because of this wide range of possible causes. So, while the improvement in treatment outcomes is very gratifying, I’m afraid that I felt that the study was of little value.

1. Is Group 2 the cases from 2010 to 2016 or 2006 to 2010?

2. there was no mention of the chemotherapy regimen. Were you treated with the same chemotherapy regimen in the two time periods? Please provide details.

3. I think we need to do propensity score matching to remove bias and compare the results.

Author Response

Dear Review, 

Thank you very much for your review of the manuscript and all critical remarks.

You are right that both changes in the diagnostics and the brachytherapy  methods lead to difficult to determine which of these changes had a greater impact on improving treatment. For this reason our most important intention in this analysis was to elaborate that all change are important but local disease advanced still remains one of the most important parameters even in patients with metastatic lymph nodes in PET.

Ad 1. Group 2 was treated from 2010 to 2016.  

Ad 2. We had mention of the chemotherapy regimen in the patients characteristic with number of courses in both groups and also relations between IIB and IIIB in the groups. All patient received the same chemiotehrapy with Cisplatyn 40mg/m² one a week during the teleradiotherapy.

Ad 3. We compare results between the groups only according to 5-year survival, because of many different parameters of diagnostic and treatment methods.  In the groups we compared the results  according to stage, histopathology, doses of radiotherapy and chemotherapy. We were looking for the most important factors which lead to higher survival rate in the groups, not between the group 1 and 2.

Reviewer 2 Report

Dear Authors,

Thank you very much for submitting your manuscript. 

Diagnosis and treatment of cervical cancer has experienced an evolution in the last decades. The topic is relevant and exciting to the field of the journal. The text is clear and easy to read. The manuscript has an excellent methodical description. The overall paper is organized and well written. The methods, the overall study design, and statistical analysis are clearly described. Discussions and the literature review are insightful and informative. The figures and tables are well presented and easy to read and understand. The presented aspects sufficiently support the conclusions. The English language seems to be good. The references are up to date and appropriate.

After reviewing the study, I have some suggestions and corrections to discuss with you:

  • The abbreviations must be explicated when they appear for the first time in the text. Please verify all the abbreviations again.
  • All Figures, Schemes and Tables should be inserted into the main text close to their first citation and must be numbered following their number of appearance.

Author Response

Dear Review, 

Thank you very much for your review of the manuscript and all critical remarks.

We explicated all abbreviations as you suggested in the text. We also inserted all Figures, Schemes and Tables  into the main text close to their first citation and verify the numbers of all.

Reviewer 3 Report

The paper addresses two critical  aspects of the cervical cancer management: the appropriate diagnosis and the optimal treatment of this cancer type. The study underlie the importance of  early detection and the superiority of certain treatments. It is an original study which brings novelty for the scientific community and for clinicians, but the manuscript has to be rewritten/resubmitted because in the present form the information will not reach the journals readership. Some basic elements of scientific manuscript writing are missing, see some examples below. 

 The manuscript have to be carefully revised, starting with the title "combined chemioradiotherapy" 

Abstract, row 16: reformulation is needed: "From this time both diagnostic and radiotherapy techniques have developed..." and "However if ther is a relationship between..."

In Introduction, it is a repetitive formulation in rows 37-38 "Cervical cancer is one of the most common cancers worldwide, and the 7th most common in Europe. It is the third most common cancer in women."

In chapter: "Even though combination therapy is superior..."rows 47-53 it is a mixture between the discussion on the benefits of the new diagnostic methods and treatment protocols

Row 59: "treated at our institution"- please nominate the institution in this chapter too

 Materials and methods: rows 76-89: the patients anonymized characteristics should be described in a table, or two distinct tables corresponding to the two arms of the study. It is nowhere mentioned Table 1 in this part of the text, although it seems that Table 1 comprise certain matching data. More details should be described about groups 1 and 2. Inclusion and exclusion criteria were not clearly mentioned. Even if it is a retrospective study, and needs no special approvals, please refer to the general ethical statements and GDPR regulations of the host institution.

Row 102: ICRU 38- the abbreviation is not explained in the text, reference 10 comprise as well the abbreviation only.

Rows 91-110 comprises the description of the group 1 treatment, not the group 1 patients characteristics. The fractions of EBRT are not mentioned, no details on the drugs/doses used in the weekly chemotherapy. Several typo such "patints"(row 108) .

Row 128: the meaning of G1, G2, G3 histopathology classification is not described

Row 131: " Most patients (n=112, 94.1%) underwent 3-dimensional conformal radiotherapy (3D-CRT)"- does this group possess certain inclusion criteria for 3D-CRT or the patients were selected randomly?

 Results: The figure captions are incomplete or very brief. The statistical analysis is accurate, bringing novel information, but the authors does not refer to the figures and tables nowhere in the text.

Discussion- the discussion on the results is comprehensive, but the brackets containing the suitable  table, figure are missing, therefore the readers cannot associate the data with the conclusions. Some golden standard references from the literature were not mentioned in this chapter. Some discussion on PET-CT and the 3D-HDR tretament superiority should be inserted in this chapter, before chapter 5 Conclusions.

The references should be carefully revised, for example in reference 3 : "Concurrent cispaltyn-base radiotherapy"; reference 9: "Cegla p,..." or reference 29: "A new paradigm changind clinical..."

Author Response

Dear Review, 

Thank you very much for your review of the manuscript and all critical remarks.

We made changes in the title, abstract and introduction according to Your  suggestion. (Rows 16,47-53,59)

In the materials and methods (Rows 76-89): we provide more details about patients and change the position of table 1, with include characteristic of patients treatment. There were no exclude criteria in the first group, all patient in stage IIB and IIIB were include to the analysis. In the second group all patient with confirm metastatic lymph node in stage IIB or IIIB were included to the analysis.

Rows 102: We explained  ICRU 38 abbreviation.

Rows 91-110: We change the name of the description, added fractions of EBRT. The information of the chemotherapy drugs with dose and number of courses in each group were included in the first place.

Row 128: We described the meaning of G1, G2, G3 histopathology classification.

Row 131: We explained the criteria of the IMRT qualification in the second group patients.

Results: We tried  to refer all figures and tables in the text top of the figures or tables. We hope that now there in more legible.

Discussion: We place the brackets with the numbers of table and figure in the discussion.

The references: We carefully revised them and we hope there is no more mistakes in this point.

The manuscript was review by the English native speaker with experience in medical English text before submission.

Round 2

Reviewer 1 Report

I very much appreciate the authors' efforts to meet the referees' requirements. Unfortunately, the authors were unable to respond to my instructions and their manuscript was not essentially different from its original version. Consequently, I regret having to recommend that this manuscript should not be accepted for publication.

Author Response

Dear Review,

Thank you very much for your second review of the manuscript and all critical remarks.

We tried to be more précised of the aim in our study and we hope that we get it right in your opinion. We answered for the revision in the best way. You wrote that “the reasons for the better outcomes with the newer treatment periods are probably a combination of a variety of factors, including diagnostic accuracy for pelvic lymph node metastases, improved accuracy of external-beam irradiation (boost irradiation with IMRT), improved brachytherapy (from 2-D to 3-D), and changes from low-dose rate to high-dose-rate” and we agree with that opinion and also wrote this in the first place in our discussion. That’s way we analyzed factors in the groups not only between these two groups of patient and we were looking for the factors independent of this changes.

Hypothesis of this study was to determined factors correlated with 5 –years survival independent of changing in diagnostic and treatment in two different group of patients, especially correlation between locally advanced of disease and survival. In the first group we didn’t know the status of the lymph nodes and other organs, but even that we improved the highest correlation between stage and tumor size and survival.  FIGO stage from 2018 put all patients with metastasis to the pelvic lymph nodes to one stage IIIC1, but we improved that even with the metastatic lymph nodes the most important factor is still locally advantage of the tumor and size of the cancer. We also show that even in patients with metastatic lymph nodes the most important factor is, not IMRT boost for the metastatic lymph node, but one again the tumor stage and size. Treatment with right does to the tumor volume, according to ICRU, is important, especially in 3D brachyteharpy. We improved that smaller doses in 3D BRT lead to lower survival, but this lower doses to the tumor was also correlated with volume of the tumor and higher doses to the organ at risk.

In the first revision You wrote ‘3. I think we need to do propensity score matching to remove bias and compare the results.”. We compare the results in the groups between the same factors like:  stages (IIB vs. IIIB), tumor size, histopathology, grading, EBRT doses, BRT doses, total radiotherapy doses and chemotherapy. Only metastasis to the lymph nose were evaluated only in the group 2, were we had knowledge about it.

We must say that we are also a little confused because in the first revision for the questions:  Are all the cited references relevant to the research?, Are the results clearly presented? the answer was “Yes”.  For the question: Are the methods adequately described, the answer  was “can be improved”, and only for the last tree it was “must be improved”. Now in the second round of review all the questions are answer:” must be improved”. 

We hope that our explanations of the aims in the study and changes we made in the manuscript will be satisfactory for you.

Reviewer 3 Report

The authors made several improvements in the paper. The quality of presentation is better, still they are several minor errors, an example from the abstract "The ami of the study was to evaluated the relationships between...",  Figure 1 and Figure 2 captures remained minimal, the four captured images should be briefly described, and nowhere in the main text these two figures are not referred.  Despite the minor errors, the conclusions are sound and they could bring novelty in the cancer patients management. 

Author Response

Dear Review,

Thank you very much for your second review of the manuscript and all critical remarks.

We made changes with you last suggestion. Thanks you very much for your opinion of this manuscript.